# Synthesis, Characterization and Broad-Spectrum Bactericidal Effects of Ammonium Methyl and Ammonium Ethyl Styrene-Based Nanoparticles

**DOI:** 10.3390/nano12162743

**Published:** 2022-08-10

**Authors:** Silvana Alfei, Debora Caviglia, Gabriella Piatti, Guendalina Zuccari, Anna Maria Schito

**Affiliations:** 1Department of Pharmacy, University of Genoa, Viale Cembrano, 16148 Genoa, Italy; 2Department of Surgical Sciences and Integrated Diagnostics (DISC), University of Genoa, Viale Benedetto XV-6, 16132 Genoa, Italy

**Keywords:** multi-drug-resistant Gram-positive and Gram-negative clinical isolates, ammonium methyl styrene copolymer, ammonium ethyl styrene homopolymer, membrane disruptors, self-forming hydrogel nanoparticles (NPs), broad-spectrum antibacterial NPs, time-killing experiments

## Abstract

Untreatable infections, growing healthcare costs, and increasing human mortality due to the rising resistance of bacteria to most of the available antibiotics are global phenomena that urgently require the discovery of new and effective antimicrobial agents. Cationic macromolecules, acting as membrane disruptors, are widely studied, and several compounds, including two styrene-based copolymers developed by us (P5 and P7), have proved to possess potent broad-spectrum antibacterial effects, regardless of the resistance profiles of the bacteria. Here, we first reported the synthesis and physicochemical characterization of new cationic nanoparticles (NPs) (**CP1** and **OP2**), obtained by polymerizing the monomers 4-ammoniummethylstyrene (4-AMSTY) and 4-ammoniumethylstyrene (4-AESTY) hydrochlorides, whose structures were designed using the cationic monomers of P5 and P7 as template compounds. The antibacterial activity of **CP1** and **OP2** was assessed against several Gram-positive and Gram-negative multi-drug resistant (MDR) pathogens, observing potent antibacterial effects for both **CP1** (MICs = 0.1–0.8 µM) and **OP2** (MICs = 0.35–2.8 µM) against most of the tested isolates. Additionally, time-killing studies carried out with **CP1** and **OP2** on different strains of the most clinically relevant MDR species demonstrated that they kill pathogens rapidly. Due to their interesting physicochemical characteristics, which could enable their mutual formulation as hydrogels, **CP1** and **OP2** could represent promising ingredients for the development of novel antibacterial dosage forms for topical applications, capable of overcoming severe infections sustained by bacteria resistant to the presently available antibiotics.

## 1. Introduction

Cationic nanosized macromolecules (cationic nanoparticles (CNPs)) including cationic dendrimers [1], quaternized carbon quantum dots (qCQDs) [2], as well as polymers and amphiphilic copolymers [3], have been and are extensively studied by experts in various sectors of microbiology. In addition to biomedicine, research related to environmental health and the food industry, especially that of food packaging, are also areas affected by CNPs for their ability to limit or inhibit bacterial growth, both in solution and on surfaces. CNPs perform well as antibacterial/bactericidal agents because they are positively charged and, by mimicking the behavior of natural cationic antimicrobial peptides (NCAPs), kill bacteria on contact, causing irreparable damage in their anionic membranes up to their disruption [1,2,3]. Particularly, following the electrostatic interactions of CNPs with the anionic constituents of the bacterial surface and their diffusion to the cytoplasmic membrane (CM), the bacterial membrane depolarizes, and progressive permeabilization occurs, leading to membrane disruption, loss of cytoplasmic content, and bacterial death [1,2,3,4,5,6,7]. Due to this particular and non-specific mechanism of action, CNPs proved fast inhibitory/killing effects, significant selectivity for bacteria, and reduced tendency to develop resistance, which represents one of the most alarming global problems, leading to untreatable infections, huge health care costs and high mortality rates [1,3,6,7,8].

Antibiotic resistance is easily acquired through several unrelated mechanisms that include, among others, mutation of targets, the action of specific enzymes and modification of cell permeability, expression of efflux pumps, etc.

In this regard, CNPs, which act upon simple contact with cell surfaces, do not need to enter bacteria and interact with specific and mutable enzymatic processes [1,3], making useless the ability of bacteria to develop resistance through genetic mutations [1,3].

To prepare homopolymers and copolymers, increasingly affective against both Gram-positive and Gram-negative pathogens, several monomers containing permanently cationic groups, in the form of tetra alkyl ammonium residues, have been synthetized and employed in reactions of polymerization to obtain cationic macromolecules [1,3,4,5,6].

Aiming to obtain CNPs that are increasingly selective for bacteria cells, monomers with primary ammonium groups, often in the form of hydrochloride salts, were considered and exploited to achieve CNPs [9,10]. Cationic macromolecules prepared with such monomers, due to the type of cationic functions, could possess properties similar to those of NCAPs containing lysine [3,4,5,6,7,11]. Interestingly, some types of CNPs possessing primary ammonium groups have outperformed their tertiary and quaternary analogs both in their antibacterial potency and in a lower level of toxicity [3,4,5,6,7,11]. Moreover, the widely recognized strategy of diluting the cationic groups of the homopolymers with uncharged blocks or randomly distributed non-cationic units deriving by the copolymerization of cationic monomers with acrylates, methacrylates, acrylamides, and methacrylamides succeeded in reducing excessive cationic charges, responsible for high toxicity, and has allowed balancing the hydrophobic and cationic properties of CNPs [11].

The selectivity of CNPs towards bacterial membranes, as well as towards tumor cells, derives mainly from the essential differences in the lipid composition of the cell membrane and in the surface constituents existing between the prokaryotic and cancer cells and the normal eukaryotic ones [8,12]. 

Typically, while the CM of normal eukaryotic cells is less anionic due to the presence of several neutral lipids, such as phosphatidylcholine, sphingomyelin, and phosphatidylethanolamine, and of a minimal fraction of negatively charged phosphatidylserine [8,13], the cell membrane of bacteria contains mainly negatively charged molecules such as phosphatidylserine, phosphatidylglycerol or cardiolipin. Therefore, bacteria more easily attract cationic molecules, such as CNPs. Additionally, while, in Gram-positive bacteria, teichoic acids impart to the cell surface net negative charges, in Gram-negative bacteria, the additional outer membrane (OM), containing phospholipids and lipopolysaccharides, imparts a strong negative charge to the cell surface [8]. Furthermore, the CM of normal eukaryotic cells, due to the presence of different classes of sterols, including cholesterol, which is absent in the cell membranes of prokaryotes, results significantly more rigid, thus protecting cells against the damaging action of cationic macromolecules [8,14].

Recently, we have reported on the synthesis, physicochemical characterization, and pharmacological properties (antibacterial and anticancer effects) of two styrene-based copolymers (P5 and P7), endowed with powerful broad-spectrum antibacterial effects, regardless of the resistance profiles of bacteria [9,10,15]. Here, we first reported the synthesis and physicochemical characterization of new CNPs (**CP1** and **OP2**), obtained by polymerizing the monomers 4-AMSTY and 4-AESTY hydrochlorides (**M1** and **M2**) (Figure 1), designed using the cationic monomers of P5 and P7 as template compounds, and by opportune structural modifications. 

Particularly, on the results from the preliminary studies of homo-polymerization and copolymerization of **M1** and **M2**, we prepared the copolymer **CP1** and the homopolymer **OP2**, whose structures were defined by spectroscopic techniques, and which were shown to possess the physicochemical properties suitable for a possible topical and/or systemic clinical application. Therefore, the antibacterial activity of **CP1** and **OP2** was assessed against several Gram-positive and Gram-negative MDR pathogens, observing potent antibacterial effects for both **CP1** and **OP2** against most of the tested isolates. Additionally, time-killing studies carried out with **CP1** and **OP2** on different strains of the most clinically relevant species of bacteria demonstrated their rapid and definitive bactericidal effects. 

## 2. Materials and Methods

### 2.1. Chemicals and Instruments

4-Chloromethylstyrene (**1a**) and all the other reagents and solvents were from Merck (formerly Sigma-Aldrich, Darmstadt, Germany) and were purified by standard procedures. Azo-bis-isobutyronitrile (AIBN) was crystallized from methanol (MeOH). 4-(2-bromoethyl)-styrene (**1b**) was prepared by a known procedure [16]. The organic solutions were dried over anhydrous magnesium sulphate and were evaporated using a rotatory evaporator operating at a reduced pressure of about 10–20 mmHg. The melting ranges of the solid compounds in this study were determined on a 360 D melting point device with a resolution of 0.1° C (MICROTECH S.R.L., Pozzuoli, Naples, Italy). The melting points and boiling points are uncorrected. The FTIR spectra were recorded as films or KBr pellets on a Perkin Elmer System 2000 instrument (PerkinElmer, Inc., Waltham, MA, USA), while ATR-FTIR analyses were carried out using a Spectrum Two FT-IR Spectrometer (PerkinElmer, Inc., Waltham, MA, USA) ^1^H and ^13^C NMR spectra were acquired on a Bruker DPX spectrometer (Bruker Italia S.r.l., Milan, Italy) at 300 and 75.5 MHz, respectively. Fully decoupled ^13^C NMR spectra were reported. Chemical shifts were reported in ppm (parts per million) units relative to the internal standard tetra-methyl-silane (TMS = 0.00 ppm), and the splitting patterns were described as follows: s (singlet), d (doublet), t (triplet), q (quartet), m (multiplet), and br (broad signal). Mass spectra were obtained with a GC-MS Ion Trap Varian Saturn 2000 instrument (Varian, Inc., Palo Alto, CA, USA; EI or CI mode; filament current: 10 mA) equipped with a DB-5MS (J&W) capillary column. Elemental analyses were performed with an EA1110 Elemental Analyzer (Fison Instruments Ltd., Farnborough, Hampshire, England). The UV-Vis spectra were acquired using a UV-Vis spectrophotometer (HP 8453, Hewlett Packard, Palo Alto, CA, USA) equipped with a 3 mL cuvette. HPLC analyses were performed on a Jasco model PU-980 instrument (JASCO Corporation, Hachioji, Tokyo, Japan), equipped with a Jasco Model UV-970/975 intelligent UV/Vis detector (JASCO Corporation, Hachioji, Tokyo, Japan) at room temperature. A constant flow rate (1 mL/min), UV detection at 254 nm, a 25 × 0.46 cm Hypersil ODS 5 mm column, and a mixture of acetonitrile/water 6/4 as an eluent were employed for the acquisitions. GC-FID analyses were performed on a Perkin Elmer Autosystem (Varian, Inc., CA, USA), using a DB-5, 30 m, diameter 0.32 mm, and a film 1 mm capillary column. Column chromatography was performed on Merck silica gel (70–230 mesh). Viscosity measurements were performed with an Ubbelhode micro viscosimeter (SI Analytics, Hattenbergstr, Germany) at 30 °C in MeOH. Dynamic Light Scattering (DLS) and Z-potential (ζ-p) determinations were performed using a Malvern Nano ZS90 light scattering apparatus (Malvern Instruments Ltd., Worcestershire, UK). Potentiometric titrations were carried out using a Hanna Micro-processor Bench pH Meter (Hanna Instruments Italia srl, Ronchi di Villafranca Padovana, Padova, Italy), which was calibrated using standard solutions at pH =4, 7, and 10 before titrations. Lyophilizations were performed using a freeze–dry system (Labconco, Kansas City, MI, USA). Thin layer chromatography (TLC) was carried out using aluminum-backed silica gel plates (Merck DC-Alufolien Kieselgel 60 F254, Merck, Washington, DC, USA), and detection of spots was made by UV light (254 nm) using a Handheld UV Lamp, LW/SW, 6W, UVGL-58 (Science Company^®^, Lakewood, CO, USA). 

### 2.2. N-[(4-Vinylphenyl) alkyl] Phthalimides ***2a*** and ***2b***

A mixture of the opportune 4-haloalkylstyrene (**1a** or **1b**) (26.6 mmol), potassium phthalimide (27.4 mmol), and dry N, N-dimethylformamide (DMF) (25 mL) was heated at 55 °C under nitrogen (N_2_) and mechanical stirring for 17 h. After the removal of the solvent at reduced pressure, the solid residue was treated with chloroform (CHCl_3_, 40 mL), filtered, and washed with CHCl_3_ (3 × 10 mL). All of the organic extracts were combined, washed with 0.2 M NaOH (15 mL), water (2 × 15 mL), and dried over anhydrous MgSO_4_. The removal of the solvent at a reduced pressure provided **2a** or **2b** as crude solids that were crystallized from MeOH.

#### 2.2.1. N-(4-Vinylbenzyl) Phthalimide (**2a**)

A yield of 81%. White crystals. Mp 107 °C; (lit.: 107–108 °C [17]). Purity 99% by HPLC. FTIR (KBr, ν, cm^−1^) 1704 (C=O), 995, 914 (CH_2_=CH). ^1^H NMR (300MHz, CDCl_3_, ppm): 4.82 (s, 2H); 5.21 (dd, 1H, *J_gem_* = 0.9 Hz; *J_cis_* = 10.9 Hz); 5.70 (dd, 1H, *J_gem_* = 0.9 Hz; *J_trans_* = 17.6 Hz); 6.66 (dd, 1H, *J_cis_* = 10.9 Hz; *J_trans_* = 17.6 Hz); 7.36 (m, 4H); 7.67–7.82 (m, 4H). ^13^C NMR (75.5 MHz, CDCl_3_, ppm): 41.31, 114.13, 123.31, 126.47, 128.84, 132.10, 133.96, 135.86, 136.31, 137.18, 167.98. GC-MS (CI, m/z, %): 264 (M^+^ +1, 100). Anal. Calcd. for C_17_H_13_NO_2_: C, 77.55; H, 4.98; N, 5.32. Found: C, 77.32; H, 5.00; N, 5.32.

#### 2.2.2. N-[2-(4-Vinylphenyl) ethyl] Phthalimide (**2b**)

A yield of 75%. White crystals. Mp 135–137 °C. Purity 98% by HPLC. FTIR (KBr, ν, cm^−1^) 1703 (C=O), 990, 907 (CH_2_=CH). ^1^H NMR (300 MHz, CDCl_3_, ppm): 2.98 (t, 2H, *J* = 7.8 Hz); 3.91 (t, 2H, *J* = 7.8 Hz); 5.20 (dd, 1H, *J_gem_* = 1.0 Hz; *J_cis_* = 10.9 Hz); 5.70 (dd, 1H, *J_gem_* = 1.0 Hz; *J_trans_* = 17.6 Hz); 6.67 (dd, 1H, *J_cis_* = 10.9 Hz; *J_trans_* = 17.6 Hz); 7.21–7.32 (m, 4H); 7.69–7.82 (m, 4H). ^13^C NMR (75.5 MHz, CDCl_3_, ppm): 34.28, 39.14, 113.42, 123.22, 126.40, 129.01, 132.06, 133.90, 136.01, 136.54, 137.63, 168.14. GC-MS (CI, m/z, %): 278 (M^+^ +1, 100). Anal. Calcd. for C_18_H_15_NO_2_: C, 77.96; H, 5.45; N, 5.05. Found: C, 77.86; H, 5.47; N, 5.04.

### 2.3. Hydrazinolysis of Phthalimides ***2a*** and ***2b***

Phthalimide **2a** or **2b** (38.3 mmol) was dissolved in 95% ethanol (EtOH, 50 mL) and treated under N_2_, and stirred at reflux with a solution of hydrazine hydrate (2.74 g, 54.7 mmol) in 95% EtOH (5 mL) for 2.5 h up to the disappearance of the reagents spot in TLC analyses (eluent benzene). After the removal of the solvent at a reduced pressure, the solid residue was treated with CHCl_3_ (50 mL) and then with 20% aqueous NaOH (50 mL). The aqueous phase was separated, extracted with CHCl_3_ (3 × 50 mL), and the extracts were combined and dried over MgSO_4_. The removal of chloroform gave the free bases **3a** (90%) or **3b** (92%) as oils which were transformed into their hydrochlorides without further purification. Anyway, the free base **3a** was also vacuum distilled and characterized. Bp 58–60 °C/1 torr. ^1^H NMR (300 MHz, CDCl_3_, ppm): 1.76 (bs, 2H); 3.84 (s, 2H); 5.22 (dd, 1H, *J_gem_* = 0.9 Hz; *J_cis_* = 10.9 Hz); 5.72 (dd, 1H, *J_gem_* = 0.9 Hz; *J_trans_* = 17.6 Hz); 6.70 (dd, 1H, *J_cis_* = 10.9 Hz; *J_trans_* = 17.6 Hz); 7.24–7.39 (m, 4H). ^13^C NMR (75.5 MHz, CDCl_3_, ppm): 140.3, 138.7, 137.8, 129.8, 128.2, 114.7, 45.20. GC-MS (ESI^+^, MeOH/H_2_O 1/1) m/z 134.0 [M + H] ^+^ (C_9_H_11_N); m/z 117.0 [M—NH_2_] ^+^ (C_9_H_9_). Anal. Calcd. For C_9_H_11_N: C, 77.96; H, 5.45; N, 5.05. Found: C, 77.86; H, 5.47; N, 5.04.

### 2.4. Synthesis of Hydrochlorides ***4a*** and ***4b***: 4-AMSTY and 4-AESTY (Monomers **M1** and **M2**)

A solution of the free amine **3a** or **3b** (25 mmol) in dry diethyl ether (500 mL) was cooled to 0 °C and treated under stirring with dry gaseous hydrochloric acid up to saturation. The white precipitate was filtered, washed with fresh ether, dried, and crystallized from 2-propanol (**M1**) or EtOH (**M2**) to provide the hydrochloride derivatives **4a** and **4b**, which were used as active monomers, namely **M1** and **M2**, in the reaction of copolymerization and homo-polymerization, respectively.

#### 2.4.1. 4-Aminomethylstyrene Hydrochloride (4-AMSTY) **4a** (**M1**) 

A yield of 89%. Mp 180 °C (dec.; 2-propanol). UV-Vis (3.6 µg/mL, MeOH): λ max 252.0 nm, ABS = 0.6192. ATR-FTIR (ν, cm^−1^) 989 and 901 (CH_2_=CH). ^1^H NMR (300 MHz, CD_3_OD, ppm): 4.11 (s, 2H); 5.29 (dd, 1H, *J_gem_* = 0.90 Hz; *J_cis_* = 10.9 Hz); 5.83 (dd, 1H, *J_gem_* = 0.90 Hz; *J_trans_* = 17.6 Hz); 6.76 (dd, 1H, *J_cis_* = 10.9 Hz; *J_trans_* = 17.6 Hz); 7.42–7.52 (m, 4H). ^13^C NMR (75.5 MHz, CD_3_OD, ppm): 44.09, 115.36, 127.91, 130.35, 133.80, 137.37, 139.93. Anal. Calcd. For C_9_H_12_ClN: C, 63.72; H, 7.13; N, 8.26; Cl, 20.90. Found: C, 63.75; H, 7.12; N, 8.22; Cl, 20.88.

#### 2.4.2. 4-Aminoethylstyrene hydrochloride (4-AESTY) **4b** (**M2**) 

A yield of 77%. Mp 210 °C (EtOH). UV-Vis (13 µg/mL, MeOH) λ max 250.0 nm, ABS = 1.238. ATR-FTIR (ν, cm^−1^) 994 and 912 (CH_2_=CH). ^1^H NMR (300 MHz, CD_3_OD, ppm) 2.92–3.03 (m, 2H); 3.12–3.23 (m, 2H); 5.21 (dd, 1H, *J_gem_* = 1.0 Hz; *J_cis_* = 10.9 Hz); 5.76 (dd, 1H, *J_gem_* = 1.0 Hz; *J_trans_* = 17.6 Hz); 6.72 (dd, 1H, *J_cis_* = 10.9 Hz; *J_trans_* = 17.6 Hz); 7.24–7.44 (m, 4H). ^13^C NMR (75.5 MHz, CD_3_OD, ppm): 34.22, 41.89, 114.08, 127.75, 130.03, 137.52, 137.70, 138.04. Anal. Calcd. For C_10_H_14_ClN: C, 65.39; H, 7.68; N, 7.63; Cl, 19.30. Found: C, 65.42; H, 7.69; N, 7.65; Cl, 19.30. 

### 2.5. Radical Solution Copolymerization of M1 with N,N-Dimethyl Acrylamide (DMAA) (**CP1**)

Degassed monomer **M1** (120.5 mg, 0.7103 mmoli), DMAA (167.3 mg, 1.6876 mmoli), MeOH (1.2 mL), and AIBN as an initiator (3.4 mg) were introduced under nitrogen in the polymerization flask and magnetically stirred. After 24 h at 60 °C, the mixture was poured into diethyl ether, and the polymer was filtered, submitted to two dissolution/precipitation cycles with MeOH/diethyl ether, and vacuum-dried at room temperature.

**CP1**: 223.9 mg, conversion yield = 77.8%. Inherent viscosity (η _inh_): 1.22 dL/g.

ATR-FTIR (KBr, ν, cm^−1^): 3500–3000 (NH_3_^+^), 2992 (CH_2_), 1607 (C=O), 838 (CH bending phenyl 1,4-disubstituted).

### 2.6. Radical Solution Polymerizations of **M2** (**OP2**)

Degassed monomer **M2** (197.2 mg, 1.0736 mmoli), H_2_O (2.5 mL), MeOH (2.0 mL), and (NH_4_)_2_S_2_O_8_ as an initiator (3.6 mg, 1.8%) were introduced under nitrogen in the polymerization flask and magnetically stirred. After 24 h at 35 °C, the mixture was poured into acetone, the polymer was filtered, submitted to two dissolution/precipitation cycles with H_2_O/MeOH 1/1 acetone, and vacuum-dried at room temperature.

**OP2**: 141.2 mg, conversion yield = 71.6%. Inherent viscosity (η _inh_): 0.46 dL/g.

ATR-FTIR (KBr, ν, cm^−1^): 3500–3000 (NH_3_^+^), 2974, 2890 (CH_2_), 1600, 1470 (stretching C=C), 828 (CH bending phenyl 1,4-disubstituted).

### 2.7. Determination of the Relative Molecular Mass (Mr) of **CP1** and **OP2**

The Mr di CP1 and OP2 was determined by using the Mark–Houwink Equation (1)
[η] = *K* × Mr*^a^*(1)
where [η] is the intrinsic viscosity of polymers, while *K* and *a* are constants [18] that have been measured and tabulated for several combinations of polymers, copolymers, solvents, and temperatures [19].

#### 2.7.1. Determination of Intrinsic Viscosity [η] of CP1 and OP2

First, the relative viscosity of **CP1** and **OP2** (η_rel_) was determined by using a Ubbelhode micro viscosimeter. Briefly, solutions of both polymers at polymer concentration (Cp) = 0.5 g/dL were prepared in MeOH, and the proper determinations were carried out at 30 °C. Particularly, the determinations consisted of measuring the outflow time of a fixed volume of the solvent (t_0_) and the outflow time of the same volume of polymers solutions (t) inserted in the micro viscosimeter, and the value of the relative viscosity was calculated using Equation (2).
η_rel_ = t/t_0_(2)

The determinations of times were made in triplicate and the results were expressed as the mean of three independent determinations ± standard deviation (SD). 

Secondly, [η] of **CP1** and of **OP2** has been computed from the values of η_rel_ consecutively using the Equations (3) and (4) [20].
η_sp_ = η_rel_ − 1(3)
where η_sp_ is the specific viscosity of **CP1** and **OP2**
[η] = η_sp_/Cp(4)
where Cp = 0.5 g/dL, i.e., the concentration of polymers used to measure their η_rel_.

### 2.8. Determination of NH_2_ Equivalents Contained in **CP1** and **OP2**

The NH_2_ content of **CP1** and **OP2**, in the form of hydrochloride, was obtained by volumetric titrations with a solution of HClO_4_ in acetic acid (AcOH), using quinaldine red as an indicator. The titrating solution was prepared and standardized with potassium acid phthalate by a slightly modified procedure previously reported [21]. The title of the solution was found to be 0.1612 N. The equivalents of NH_2_ contained in **CP1** and **OP2** were determined by exactly weighting the samples of **CP1** (24.2 mg) or **OP2** (31.4 mg) with a microbalance dissolving them in AcOH (5 mL), treating the obtained solutions with 2 mL of mercury acetate (1.5 g) in AcOH (25 mL), adding few drops of a solution of quinaldine red (100 mg) in AcOH (25 mL) and titrating with the standardized solution of HClO_4_ in AcOH, using a calibrated burette with needle valve. Very sharp end points were detected by observing the disappearance of the red color. Standardizations and titrations were made in triplicate, and the results were reported as means of three independent experiments ± SD and were expressed both as µequiv. NH_2_/µmol and mequiv. NH_2_/g. 

### 2.9. Dynamic Light Scattering (DLS) Analysis

The particle size (in nm), polydispersity index (PDI), and zeta potential (ζ-p) (mV) of **CP1** and **OP2** were measured at 25 °C, at a scattering angle of 90° in m-Q water by using a Malvern Nano ZS90 light scattering apparatus (Malvern Instruments Ltd., Worcestershire, UK).

Solutions of samples in m-Q water were diluted to final concentrations to have 250–600 kcps. The Ζ-p value of **CP1** and **OP2** was recorded with the same apparatus. The results from these experiments were presented as the mean of three independent determinations, made of ten runs, each one ± SD. Concerning the particle size distribution, intensity-based results were reported.

### 2.10. Scanning Electron Microscopy (SEM)

The sample was fixed on aluminum pin stubs and sputter-coated with a gold layer of 30 mA for 1 min to improve the conductivity, and an accelerating voltage of 20 kV was used for the sample’s examination. The micrographs were recorded digitally using a DISS 5 digital image acquisition system (Point Electronic GmbH, Halle, Germany). 

### 2.11. Potentiometric Titration of **CP1** and **OP2** NPs

Potentiometric titrations were performed at room temperature, and the titration curves of **CP1** and **OP2** were obtained. Exact amounts of **CP1** and **OP2** (25.5 mg **CP1** and 11.0 mg **OP2**) were dissolved in 30 mL of Milli-Q water (m-Q), then were treated under magnetic stirring with a standard 0.1 N NaOH aqueous solution [3.0 mL, pH = 10.48 (**CP1**) and 10.62 (**OP2**)]. The solutions were potentiometrically titrated under stirring by adding 0.2 mL aliquots of a standard 0.1 N HCl aqueous solution, up to 3.0 mL, 0.5 mL, up to 6.0 mL, and finally 1.0 mL up to a total of 10.0 mL and measuring the pH values of the obtained solutions [21]. Titrations were made in triplicate, and the measurements were reported as mean ± SD.

### 2.12. Microbiology

#### 2.12.1. Microorganisms

A total of 28 strains of different species of Gram-positive and Gram-negative bacteria were utilized in this study to assess the antibacterial effects of **CP1** and **OP2**. All were clinical isolates from human specimens and were identified by matrix-assisted laser desorption/ionization time-of-flight (MALDI-TOF) mass spectrometric technique (Biomerieux, Firenze, Italy) or by VITEK^®^ 2 (Biomerieux, Firenze, Italy). Of the tested pathogens, 12 were Gram-positive organisms. Particularly, 6 strains belonged to the *Enterococcus* genus (3 *Enterococcus faecalis* resistant to vancomycin (VRE)*,* and 3 *E. faecium* VRE, of which one was resistant also to teicoplanin). Six strains pertained to the *Staphylococcus* genus, including 3 methicillin-resistant *S. auresus* (MRSA) and three methicillin-resistant *S. epidermidis* (MRSE), two of which were also resistant to linezolid Concerning the 16 Gram-negative isolates, 9 strains were *Enterobacteriaceae*: 3 *Escherichia coli* (two producing carbapanemases of the KPC family and one those of NDM family), 3 *Klebsiella aerogenes* (2 not β-lactamases producing carbapenems- resistant isolates and 1 producing carbapenemases of KPC family), and 3 group A carbapenemase-producing *K. pneumoniae*. The last seven Gram-negative isolates were MDR bacteria of non-fermenting species, including 3 *Pseudomonas aeruginosa* (one strain was isolated from patients with cystic fibrosis, one was a colistin-resistant isolate, and one was a pyomelanin-producing strain), 2 *Acinetobacter baumannii,* and 2 *Stenotrophomonas maltophylia.*

#### 2.12.2. Determination of the Antibacterial Properties of **CP1** and **OP2**

##### Determination of the Minimal Inhibitory Concentrations (MICs)

The antibacterial effects of **CP1** and **OP2** on the 28 pathogens previously described were assessed by determining their Minimal Inhibitory Concentrations (MICs) following the microdilution procedures detailed by the European Committee on Antimicrobial Susceptibility Testing (EUCAST) [22]. 

Briefly, as also described in our previous works [23], overnight cultures of bacteria were diluted to obtain a standardized inoculum of 1.5 × 10^8^ CFU/mL. Appropriate aliquots of each suspension were added to 96-well microplates containing the same volumes of serial 2-fold dilutions (ranging from 128 to 1 μg/mL) of the cationic polymers (**CP1** or **OP2**) to achieve a final concentration of about 5 × 10^5^ cells/mL. The plates were incubated at 37 °C for 24 h, and then the MICs were read, observing where in the 96-well microplates the bacteria growth was inhibited. Particularly, the concentration of the first well in the series of wells at decreasing concentrations of the samples, where no bacterial growth was observed, was recorded as the MIC. DMSO not containing the tested substances was used as a control to verify the absence of antibacterial activity of the solvent used for the experiments. All MICs were obtained at least in triplicate, and the results were expressed by reporting the modal value, that is, the value that has been observed most frequently. In case of equivocal or unclear results, more than three determinations of MICs were carried out.

##### Determination of Minimal Bactericidal Concentrations (MBCs)

The MBC has been defined as the lowest concentration of a drug that results in killing 99.9% of the bacteria being tested [24]. 

The MBCs of **CP1** and **OP2** on the 28 pathogens were determined by subculturing the broths used for MIC determination. Briefly, 10 μL of the culture broths of the wells corresponding to the MIC and to higher MIC concentrations were plated onto fresh MH agar plates and further incubated at 37 °C overnight. The highest dilution that yielded no bacterial growth on the agar plates was taken as MBC. All of the tests were performed at least in triplicate, and the results were expressed as the mode. 

#### 2.12.3. Time-Kill Experiments

Killing curve assays for **CP1** and **OP2** were performed on various MDR isolates of *S. aureus, E. coli,* and *P. aeruginosa*, as previously reported [25]. The experiments were performed over 24 h at concentrations 4 times the MICs and repeated until reproducibly was achieved, and the curves shown were those whose trend has been observed most frequently.

## 3. Results and Discussion

### 3.1. Synthesis and Spectrophotometric Characterization of Monomers **M1** and **M2**

#### The Design of **M1** and **M2** Structure

With the aim of creating new nanosized cationic macromolecules with antibacterial activity and inspired by the excellent biological results that we have previously obtained with two polystyrene-based copolymers, P5 and P7 [9,10,15], we decided to synthetize the styrene monomers **M1** and **M2**. Among the previously reported potent antibacterial agents P5 and P7, P7 having cationic benzyl ammonium groups in place of the butyl ammonium ones of P5 was the most active, but the synthesis of the monomer necessary to prepare P7 is very complex and long. So, to obtain polymers with cationic groups similar to those of P7 but obtainable from easier-to-prepare monomers, we shortened the alkyl chain of four carbon atoms present in the monomer that provided P5. To this end, we thought to prepare benzyl ammonium and ethyl ammonium styrene derivatives. Monomers **M1** and **M2** were synthesized according to Figure 1 [26].

The starting material, compound **1a**, was purchased as described in Section 2.1, while compound **1b** was prepared by a described route, starting with the acylation of 2-bromoethylbenzene, to yield the acetyl derivative, whose carbonyl group was reduced to a hydroxyl group (OH) using NaBH_4_. Upon the elimination of the OH group by distillation at 200 °C, the vinyl group and, therefore, compound **1b** were obtained [16]. The reaction of **1a** or **1b** with potassium phthalimide in DMF at 55 °C provided the Gabriel adducts **2a** and **2b,** which were purified and characterized before submission to hydrazinolysis. After hydrazinolysis, by treating the Gabriel adducts with hydrazine in EtOH at 95°C, under heating, the obtained amino-methyl-styrene **3a** was purified by distillation under a vacuum and completely characterized before transformation into the corresponding hydrochloride salt (**4a**), renamed monomer **M1**. Differently, the free amine **3b** was promptly transformed into the corresponding hydrochloride salt (**4b**), renamed monomer **M2**, for better purification and easy storage. The ATR-FTIR spectrum of both monomers (**M1** and **M2**) showed the typical vinyl double bond bands at 905 and 990 cm^−1^ (**M1**) and at 912 and 994 cm^−1^(**M2**) (Figure 2a,b, respectively). Weak bands at 2830 and 2952 cm^−1^ (CH stretching of methylene group) and at 3006 and 3086 cm^−1^(CH stretching phenyl ring) were observed to come out from the typical large band of the hydrochloride in the spectrum of **M1**, where a weak but sharp band related to the free N-H stretching vibration, was detected at 3274 cm^−1^ (Figure 2a). Additionally, aromatic C=C stretching bands were visible in the spectrum of **M1** at 1590 and 1455 cm^−1^ (Figure 2a).

Such bands were also observable in the spectrum of **M2** at 1470 and 1600 cm^−1^, where bands belonging to the CH stretching of methylene groups were detected at 2887 and 2972 cm^−1^, while in the region 3200–2500 cm^−1^was detected, the broad but intense band of bound NH stretching vibration (Figure 2b). In both spectra, aromatic overtones were observable in the region 1700–2000 cm^−1^ (Figure 2a,b). The ^1^H NMR spectrum of **M1** showed a singlet signal at 4.11 ppm (-CH_2_-) (Figure 3a), while that of **M2** revealed two very close multiplets at 2.92–3.03 and at 3.12–3.23 ppm (CH_2_-CH_2_) (Figure 3b). The protons of the NH_3_^+^ groups were not detected, as these undergo exchange in the working solvent. The vinyl system provided the typical signal consisting of two double doublets, whose integrals denoted one proton for doublet, associated with a quartet (1 H) in the range 5.20–6.80 for both monomers (Figure 3a,b). The *p*-di-substituted aromatic systems provided a multiplet centered at 7.27 ppm, whose integral denoted four protons (Figure 3a,b).

The ^13^C NMR spectrum of **M1** presented seven signals, plus the multiplet signal close to 50 ppm, belonging to CD_3_OD used as a solvent for acquiring the spectrum. Particularly, one signal for the carbons of the methylene (44.09 ppm). Two signals for the vinyl system (115.36 and 137.37 ppm), two intense signals related to the four aromatic carbon atoms (127.91 and 130.35 ppm), and two signals of low intensity relating to quaternary aromatic carbons (133.80 and 139.93) were also detectable (Figure 4a). The ^13^C NMR spectrum of **M2** presented instead eight signals, six of them very similar to those of **M1**, for the vinyl system, the four aromatic carbon atoms, and the two quaternary aromatic carbons. Two signals in place of one were detectable for the two methylene groups and were observed at 34.22 and 41.89 ppm (Figure 4b). As expected, due to the presence of the same chromophore in both **M1** and **M2**, the UV-Vis spectra of the two monomers acquired in MeOH were practically identical, and maxima of absorbance at λ max = 252 nm and 250 nm were observed for **M1** and **M2**, respectively (Figure 5).

### 3.2. Radical Polymerizations in Solution 

Cationic polymers are macromolecular architectures extensively studied as antimicrobial agents because, as they are similar to natural and synthetic antimicrobial peptides (NAMPs and SAMPs), they act by interacting with the negative surface of bacteria, damaging their membranes, and causing their rapid death [3].

Additionally, random or block copolymers can be easily synthesized, merging uncharged comonomers with cationic monomers, thus obtaining macromolecules having charged and not charged moieties along the polymer backbone [8]. In this way, the amphiphilic character and hydrophobic content are coupled and strongly correlate with antimicrobial activity and the selectivity of macromolecules [8]. In this context, the CNPs (P5 and P7) obtained by us copolymerizing styrene-based monomers in the form of ammonium salts (M5 and M7) with DMAA have shown powerful ultra-broad-spectrum bactericidal effects against several species of MDR pathogens, P7 being even more potent than P5 [9,10]. Here, once the structures of the new **M1** and **M2** monomers were designed, following the previously explained reasons, and after successfully synthetizing and characterizing them, preliminary radical polymerization experiments were carried out. Such investigations showed that, while **M1** copolymerized easily with DMAA, providing CNPs in high conversion yield, **M2** did not, yielding materials with non-nanosized dimensions with unwanted physicochemical properties and with a very low conversion yield (19%). Therefore, in the present work, to structurally optimize the obtainable CNPs, **M1** was copolymerized with DMAA in MeOH using AIBN as a radical initiator at 60 °C, achieving the random copolymer **CP1** with a conversion of 78% (Figure 2a). On the contrary, **M2** gave good results when homopolymerized in an aqueous medium (H_2_O/MeOH) at a lower temperature (35 °C) and using ammonium persulfate as an initiator (Figure 2b), affording homopolymer **OP2**, with a conversion of 72%.

The experimental data of the polymerizations have been reported in Table 1.

**CP1** and **OP2** were purified by repeated cycles of dissolution/precipitation using MeOH as solvent and Et_2_O as non-solvent (**CP1**), H_2_O/MeOH 1/1 as a solvent, and acetone as a non-solvent (**OP2**).

#### Spectroscopic Characterization of **CP1** and **OP2**

In the ATR-FTIR spectra of **CP1** (Figure 6a), broad bands, typical of copolymers made of long polymer chains and having high molecular weight, were observable. Particularly, a very broad band in the range of 3500–3000 cm^−1^ due to the NH_3_^+^ groups deriving from **M1** was visible, a very weak band belonging to the methylene groups was detectable, while the contribution of DMAA was confirmed by the intense band at 1607 cm^−1^. 

In the ATR-FTIR spectra of **OP2** (Figure 6b), sharper bands were observable, indicating the presence of shorter polymer chains as also evidenced by its lower viscosity and Mr respect to **CP1**. Particularly, in the range of 3500–3000 cm^−1^, a band similar to that observable in Figure 4a, due to the NH_3_^+^ groups, was visible, while intense bands at 2974 and 2890 cm^−1^ indicated that the methylene groups were detected. Overtones in the range of 2000–1700 cm^−1^ confirmed the presence of phenyl rings, while the contribution of DMAA was absent, as expected. **OP2** was well soluble in water, MeOH, DMSO, and DMF while insoluble in all other organic solvents, including petroleum ether, diethyl ether, dichloromethane, chloroform, and acetone. Interestingly, **CP1**, at opportune concentrations, swelled in certain solvents, including DMSO and water, providing interesting hydrogels, without the addition of any additive, thinkable as future self-forming formulations for topical administration of **CP1**.

### 3.3. Determination of the Relative Molecular Mass (Mr) of **CP1** and **OP2**

The Mr di **CP1** and **OP2** were determined by using the Mark–Houwink Equation (1) reported in Section 2.7, which describes the dependence of the intrinsic viscosity [η] of a polymer on its relative molecular mass (molecular weight, Mr). *K* and *a* in the equation are constants whose values depend on the nature of the polymer and solvent, as well as on the temperature used for the determination of the value of [η] for the polymer under consideration [18]. Such constants have been measured and tabulated for several combinations of polymers, copolymers, solvents, and temperatures [19]. In this regard, to estimate the Mr of **CP1** and **OP2**, which are both styrene-based, cationic, and hydrophilic macromolecules, and whose [η] were determined in a hydrophilic solvent (MeOH) at 30 °C, we retained compatible to use the tabulated values of *a* and *k* previously measured for a similar polymer (poly-2-vynilpyridine) in a hydrophilic solvent (water) at the same temperature (30 °C). Once we selected the values of *a* and *k* to calculate the Mr of **CP1** and **OP2**, we determined their [η]. To this end, the relative viscosity of **CP1** and **OP2** (η_rel_) was first determined at 30 °C by using a Ubbelhode micro viscosimeter and solutions of polymers in MeOH at Cp = 0.5 g/dL as detailed in Section 2.7.1. Secondly, the values of [η] have been computed for both **CP1** and **OP2** from the values of η_rel_ using consecutively the Equations (3) and (4) [20] (Section 2.7.1). Table 2 summarizes the values of η _rel_ measured for **CP1** and **OP2**, the related η _sp_ and [η], the used values of *a* and *K*, and the resulting Mr obtained by the Mark–Houwink Equation (1).

### 3.4. Determinations of NH_2_ Content of **CP1** and **OP2**

To determine the NH_2_ content of **CP1** and **OP2** and have evidence of their charge density, we carried out the titration of amine hydrochlorides with HClO_4_ solution in acetic acid (AcOH) in the presence of mercuric acetate and quinaldine red as an indicator, as previously reported [21]. The method is cheap and fast, and its accuracy has been secured by a sharp endpoint of the titration, while its reliability has been demonstrated by the reproducibility of the results (Table 3). 

### 3.5. Particle Size, ζ-p and PDI of **CP1** and **OP2**

The hydrodynamic size (diameter) (Z-AVE, nm) and polydispersity index (PDI) of **CP1** and **OP2** NPs were determined by DLS analysis to assess the dimensions of particles and how much their dimensions are uniform. Additionally, ζ-p measurements were carried out to determine their surface charge. Figure 7 shows the representative particles size (Figure 7a,b) and ζ-p distributions (Figure 7c,d) of **CP1** and **OP2**, respectively, while in Table 4, we have reported the average size (Z-AVE, nm) and the mean value of PDI ± SD, which correspond to the mean of three measurements made of ten runs each one. Additionally, Table 4 also contains the mean ± SD of the values of ζ-p obtained by one measurement made of 12 runs. Particularly, in Figure 7a, the three size distributions obtained by the three measurements carried out on **CP1** have been reported overlapped, while the legend only refers to one of the three size distributions, which contributed to the Z-ave reported in Table 4 for **CP1**.

The mean particle size was very high for **CP1** NPs (833 nm) and significantly lower for **OP2** (163 nm), while the PDI values were similar. These results are in accordance with the literature data, reporting that the molecular weight of polymers increases at the growth of particle diameter and vice versa [27]. So, it was expectable that **CP1**, whose Mr has been estimated to be around 157,000, would have proved an average particle size remarkably higher than that of **OP2**, having an Mr of around 45,000. We remember that, in the case of an administration of nano-formulations, the size of NPs strongly influences its distribution, cytotoxicity, targeting ability [28,29], and biomedical applications require sizes lower than 200 nm, with an optimal of 100–200 nm [29]. Hence, due to the small dimension of its particles, **OP2**, which has also been demonstrated to be highly water soluble, could be considered suitable for systemic administration since NPs with a size of about 100 nm, as that determined for **OP2**, have a larger surface-area-to-volume ratio and are more effective and faster. On the contrary, the large dimension of particles of **CP1** causes it to be unsuitable for systemic administration since, as widely reported, particles larger than 200 nm tend to activate the lymphatic system and are removed from circulation quicker [30]. Anyway, we did not consider this a limitation for the biomedical application of **CP1** developed by us, since due to its capability to self-forming gels, it may be formulated as an antibacterial hydrogel for topical administration to treat skin and soft tissues infections sustained by MDR pathogens. Although slightly lower than the value of ζ-p determined for the copolymer P5 previously reported by us [9], as expected considering data obtained by volumetric titrations, concerning the content of ammonium groups, both **CP1** and **OP2** proved high positive ζ-p values (+31 and +27 mV, respectively), which are welcome for the development of highly effective nano-formulations. In fact, based on the studies published so far, the internalization of CNPs, such as those developed in this study, is more efficient than that of neutral and anionic NPs [31,32,33,34,35,36]. Notably, it was found that after electrostatic interaction with anionic components of cells membrane as phospholipids, CNPs can be internalized by several mechanisms, including pore formation, micropinocytosis, as well as clathrin- and dynamin-dependent endocytosis [36].

Finally, the PDI values of **CP1** and **OP2** were sufficiently low, thus indicating high stability of the water solutions of **OP2** and of hydrogels preparable with **CP1**.

### 3.6. Scanning Electron Microscopy (SEM)

To confirm the hydrodynamic size and PDI values obtained by the DLS analyses and have an idea of the morphology of the particles herein prepared, SEM analyses were carried out on the lyophilized **CP1**, as a representative example of both polymers. An SEM micrograph of **CP1** is shown in Figure 8.

The SEM image of lyophilized cationic **CP1** revealed a spherical morphology, a narrow polydispersity, and particles with diameters slightly under a micron, thus confirming the DLS data.

### 3.7. Potentiometric Titration of **CP1** and **OP2** NPs

Both **CP1** and **OP2** have neither quaternary nor permanently protonated ammonium groups but possess reversibly protonable primary amine groups, whose protonation depends on the pH value of the environment. Due to the most recognized mechanism of action of antibacterial NAMP, SAMP, as well as dendrimers and polymers possessing ammine groups, protonation is essential for having a positively charged surface, which is crucial for interacting with the surface of bacteria and for providing significant membrane disrupting effects [1,3]. Therefore, to speculate on a future formulation of **CP1** and **OP2** as antibacterial hydrogels for topical administration, we thought it crucial to know the pH values at which they can be protonated and, mainly, if they would be protonated at the pH of the skin (pH < 5). To obtain this information, potentiometric titrations of **CP1** and **OP2** were carried out as previously described [37]. The titration curves were obtained by graphing the measured pH values vs. the aliquots of HCl 0.1 N added (Figure 9a). Subsequently, from the titration data, the dpH/dV values were computed and reported in the graph vs. those of the corresponding volumes of HCl 0.1N, thus obtaining the first derivative lines of the titration curves (Figure 9b). Additionally, the potentiometric titrations of **CP1** and **OP2** allowed to titrate the NH_2_ groups of both polymers and so to further determine the NH_2_ content in **CP1** and **OP2** NPs already estimated by volumetric titration (Section 3.4, Table 3).

As observable in Figure 9a, the titration curves of both **CP1** and **OP2** were very similar, showing a very high buffer capacity up to the adding of 6–7 mL HCl 0.1 N, while the so-called jump in pH (titration point) was visible for both polymers when a volume of 0.1 N HCl of 7.8 and 8 mL respectively was added. Particularly, by determining the first derivatives of the titration curves, whose maxima represent the titration end points (or the various phases of the protonation process), we observed that, although partial protonation started at pH values of 7–9, both polymers were completely protonated at pH values of about 3.5 (Figure 9b). These results established that both **CP1** and **OP2** at the pH of skin should be sufficiently protonated, as desired, thus favoring electrostatic interactions with the negatively charged surface of bacteria and boosting up their antibacterial effects. Interestingly, as evidenced by the following results concerning the antibacterial and bactericidal effects of **CP1** and **OP2**, both types of CNPs resulted in being very potent against all Gram-positive bacteria and almost all the Gram-negative isolates herein considered when in MH media. Concerning this, since the pH of MH is about 7, a value that allows only a partial protonation of the polymers, as shown in Figure 9b, it can be speculated that the already potent antibacterial effects observed in vitro for **CP1** and **OP2** are only an underestimation of those that they could exert in vivo at the pH of the skin that permits a higher degree of protonation. Table 5 collect the results concerning the content of the NH_2_ group in **CP1** and **OP2**, which confirmed the values obtained by volumetric titrations previously reported, with a minimal error of 1.3% and 0.3%, respectively.

### 3.8. Antibacterial Properties

#### 3.8.1. Antimicrobial Activity of **CP1** and of **OP2**

The MIC values for **CP1** and **OP2** were obtained by analyzing a total of 28 strains of clinical and multi-resistant origin, including both Gram-positive and Gram-negative species responsible for difficult-to treat-infections [38]. Even if the values of MIC > 128 µg/mL corresponded to very low µM concentrations of both **CP1** and **OP2** when the bacteria were not inhibited at 128 µg/mL, higher concentrations were not considered, and the MICs were expressed as greater than 128 µg/mL. In this regard, **CP1** displayed MICs > 128 µg/mL only against Gram-negative bacteria in five cases out of 28 (one KPC-producing *E. coli,* one KPC-producing *K. pneumoniae,* one *A. baumanni*, and two *S. maltophilia*), the highest MICs of **OP2** corresponded to 64 µg/mL. However, since traditional antibiotics and **CP1** and **OP2** themselves are characterized by very different molecular weights, to compare their antibacterial activity, we evaluated more correctly to consider the MIC values expressed as micromolar concentrations (µM). Indeed, the µM concentrations provide how many equivalents (which are comparable magnitudes) of the tested substance are necessary to achieve inhibition on a specific isolate. Overall, both compounds provided remarkably interesting results, both against Gram-positive and Gram-negative species (0.1–2.8 μM). **CP1** was found to be the compound that, in some cases, showed the lowest MICs (0.1–0.8 μM). Both **CP1** and **OP2** were particularly effective against all the MRSE isolates, especially against the two strains also resistant to linezolid (0.1 µM for **CP1** and 0.35 µM for **OP2**), while slightly higher MICs were observed against all other Gram-positive bacteria. Going into detail, **CP1**, except for MRSA 187 and VRE *E. faecalis* 450 (MICs = 0.8 µM), displayed MICs = 0.4 µM against all enterococci and isolates of the *S. aureus* species, while **OP2** displayed MICs = 0.7 µM against all MRSA and *E. faecium* isolates, and MICs = 1.4 µM against all VRE *E. faecalis.* Very low MICs (0.4 and 0.8 µM) were observed for **CP1** also on six out of nine isolates of different Gram-negative species of the *Enterobacteriaceae* family. On the contrary, **CP1** showed weak activity on non-fermenting Gram-negative bacteria, such as *A. baumannii* and *S. maltophilia*. However, it displayed MICs = 0.4–0.8 µM against *P. aeruginosa*, including a colistin-resistant strain (MIC = 0.8 µM). The lowest MIC for the genus *Pseudomonas* was observed against a strain producer of pyomelanin, confirming the high susceptibility of isolates with this negatively charged pigment to cationic macromolecules, already reported by us [25]. Interestingly, while less potent than **CP1** against staphylococci, enterococci, and all the *Enterobacteriaceae* except for *K. aerogenes* 500 (MICs = 1.4 µM vs. MICs 0.4–0.8 µM), with the exclusion of the strain of *P. aeruginosa* resistant to colistin (MICs = 1.4 µM), **OP2** was more potent than **CP1** against all non-fermenting Gram-negative isolates considered in this study, proving good activity (MICs = 0.7 µM) just against strains on which **CP1** was only weakly active (*A. baumannii* and *S. maltophilia*). In this regard, it could be thought to merge **CP1** and **OP2** in a single dosage form to obtain a new antibacterial formulation more powerful than the separate molecules and with a wider spectrum of activity. To this end, since we have observed that **CP1**, at adequate concentrations, produces hydrogels, it can be used as a gelling agent to formulate **OP2**. The exploration of such a possibility could be the subject of our future studies. Overall, the results reported here on the antibacterial activity of **CP1** (MIC = 0.1–0.8 µM) and **OP2** (MIC = 0.35–2.8 µM) evidenced that both polymers were remarkably more potent than P5 previously reported by us [9], which differs from **CP1** and **OP2** for the length of the carbon spacer between the phenyl ring and the ammonium group, responsible for the antibacterial effects. Practically, this study demonstrated that the reduction in the number of methylene groups of the alkyl chain used as a spacer and the consequent approach of the NH_3_ ^+^ group to the aromatic ring in the active monomers used to prepare **CP1** and **OP2**, translate sinto the enhancement of their antibacterial properties. We can think that with these modifications, we have obtained NPs whose structures, compared to P5, can interact electrostatically better with the bacterial surface, which is an essential step for the effectiveness of cationic antimicrobial compounds since they act as membrane disruptors.

In the fourth column of Table 6, we have reported the MICs of the antibiotics commonly used against Gram-positive and Gram-negative strains tested in this study to compare their antibacterial effects with those of **CP1** and **OP2**. Considering the MICs expressed in micromolar concentrations (µM), for the reasons above reported, data in bold in Table 6 established that, at least in vitro, the **CP1** and **OP2** NPs developed by us are remarkably more efficient than antibiotics that are no longer effective on the bacteria considered here.

#### 3.8.2. Time-Killing Curves

To evaluate the possible bactericidal effects of **CP1** and **OP2**, MBCs were determined on all strains reported in Table 6, obtaining the values of concentrations at max 2-fold the MICs (not reported results), thus establishing a possible bactericidal activity for both samples. Therefore, to evaluate the possible bactericidal activity of **CP1** and **OP2**, time–kill experiments were performed with two NPs at concentrations equal to 4 × MIC on different strains of *P. aeruginosa*, *E. coli*, and *S. aureus*. As depicted in Figure 10, showing the most representative curves obtained for each species, **CP1** possessed an extremely strong bactericidal effect against all the pathogens tested, regardless of their resistance pattern, since a rapid decrease of >4 logs in the original cell number was evident already after 2 h of exposure for an MRSA, for an MDR strain of *P. aeruginosa* isolated from a patient with cystic fibrosis and for a colistin-resistant *P. aeruginosa.* The same massive reduction in the initial inoculum was also observed after 3 h against an NDMs-producing *E. coli* and was maintained after incubations of 24 h (Figure 10a).

Even more rapid were the bactericidal effects of **OP2** against all of the isolates tested and especially on *S. aureus,* since a decrease of ≥4 logs in the original cell number was evident already after 1 h of exposure. No regrowth was noted after 24 h of incubation also with **OP2** for all the three species tested (Figure 10b). Interestingly, this behavior is different from that already observed for cationic bactericidal peptides, such as colistin [39], some dendrimers [6,40], and polymers, which prove the ability to kill bacteria very rapidly, simply on contact, respectively, after an exposure of 5 min [39], 1 h [40], and 1–4 h [6,41]) but for which a regrowth was then observed after 24 h. In contrast, the bactericidal behavior observed here for **CP1** and **OP2**, devoid of regrowth at 24 h, is similar to that of the P5 copolymer (with longer linkers bearing the ammonium groups), the styrene P7 copolymer (with benzyl ammonium groups), and pyrazoles-containing NPs recently reported by us [9,10,23].

## 4. Conclusions and Future Perspectives

In this study, by means of rational structural modifications on two styrene-based monomers (**M5** and **M7**), which provided cationic copolymers with potent broad-spectrum bactericidal properties, we designed and synthetized two new monomers 4-AMSTY (**M1**) and 4-AESTY (**M2**) that differ each other in the length of the carbon chain linking the primary positively charged ammonium group and the styrene ring. Upon a complete physicochemical characterization and preliminary tests of radical polymerization in solution, to have optimized cationic macromolecules, we copolymerized **M1** with DMAA and omo-polymerized **M2** in proper conditions obtaining new CNPs, containing 4-AMSTY (**CP1**) and 4-AESTY (**OP2**) hydrochloride moieties, respectively, which differs a lot in term of particle size, intrinsic viscosity, and molecular weight. Such differences greatly influenced the behavior of the two polymers in a water solution, providing very interesting outcomes for their future formulation and possible topical administration as therapeutics to treat skin infections. On the contrary, despite their structural and physicochemical differences, the antibacterial and bactericidal properties of **CP1** and **OP2**, assessed by determining their MICs and by performing time–killing experiments, were comparable. The lowest values of MIC observed against the main families of tested Gram-positive and Gram-negative species have been summarized in Figure 11. 

According to what was displayed graphically, **CP1** was extremely potent against *S. epidermidis* (MICs = 0.1 µM), remarkably and equally potent on enterococci, staphylococci *Enterobacteriaceae* and *P. aeruginosa* (MICs = 0.4 µM), while less potent against *A. baumannii* (MICs = 0.8 µM), and weakly active towards *S. malthophilia* (MICs > 0.8 µM). On the contrary, just against these latter species, weakly inhibited by **CP1**, **OP2** was very well performant, proving very low MICs (0.7 µM). However, on other species, **OP2** was only slightly less potent than **CP1**, displaying MICs in the range of 0.35–1.4 µM. 

Figure 12 shows instead the differences in the values of Log_10_(CFU/mL), observed against an NDM-producing *E. coli*, a colistin-resistant isolate of *P. aeruginosa*, one *P. aeruginosa* strain isolated from a patient with cystic fibrosis and against an MRSA isolate, in the absence and in the presence of **CP1** and **OP2**. According to the graph, both samples are highly bactericidal, being able to reduce even more than 99.9% of the colonies of all the species tested. Dose-dependent cytotoxicity experiments on normal human fibroblast are undergoing to assess the feasibility of a possible clinical administration of both **CP1** and **OP2** as bactericidal NPs for topical administration to treat skin infections. The early results from such experiments evidenced LD_50_ values from which values of selectivity indices from good to very high (1.3–10.6 for **CP1** and 1.4–5.7 for **OP2**) can be calculated. On these promising data, based on the structural properties of **CP1,** which has been shown to self-form hydrogels when dispersed in water without any additive, it could be formulated as a bactericidal gel for topical treatment of skin and soft tissues infections sustained by MDR pathogens. Additionally, we are seriously thinking of using **CP1** as a gelling agent to formulate **OP2** as a hydrogel and simultaneously extend the spectrum of action of both **CP1** and **OP2**.

## Data Availability

All necessary data are comprised in this manuscript.

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
