# Peer review of "Synthesis, Characterization and Broad-Spectrum Bactericidal Effects of Ammonium Methyl and Ammonium Ethyl Styrene-Based Nanoparticles"

_nanomaterials, 2022, doi:10.3390/nano12162743_

Round 1

Reviewer 1 Report

The authors developed new cationic nanoparticles (NPs) which have broad-spectrum bactericidal effects, CP1 and OP2, which were synthesized based on two styrene-based copolymers P5 and P7 designed by the authors previous reports. The research presented physicochemical characterization of NPs to enable their mutual formulations as hydrogels for the development of antibacterial biomaterial and the new NPs were demonstrated to have antibacterial activities against several Gram-positive and Gram-negative multi-drug resistant pathogens. The research was properly designed and well conducted. The information is important and will contribute to the research field including biomaterials and medicine. Several comments should be addressed, which will strengthen the manuscript.

Questions

1.       The author explained that “some types of CNPs possessing primary ammonium groups have outperformed their tertiary and quaternary analogues in a lower level of toxicity” and “both CP1 and OP2 have neither quaternary nor permanently protonated ammonium groups but possess reversibly protonable primary amine groups”. This explanation can be interpreted as CP1 and OP2 having a low level of toxicity. However, there is a possibility that CP1 and OP2 could have toxicity to normal cells, as can be seen in the previously report which showed P5 and P7 exhibited to reduce NB cell viability in a concentration dependency. In addition, the authors also mentioned that dose dependent cytotoxicity experiments on human normal fibroblasts are undergoing to assess the possibility of CP1 and OP2 for clinical application in the discussion of this study. I agree with their discussion. However, cytotoxicity experiments are required in this study, not in the future, because the authors are seriously considering the usage of CP1 and OP2 as antibacterial biomaterials.

2.       The authors performed to the experiments of MICs and MBCs and the results showed remarkable antibacterial effects of CP1 and OP2. How about biofilm formation? I would like to know about the effects of them against biofilm formation. 

3.       The results of all MICs in this study were expressed reporting the modal value. To objectively prove the reliability of data, the authors should show all data of measured value including the measurements which were triplicate as a supplemental data.

Author Response

The authors developed new cationic nanoparticles (NPs) which have broad-spectrum bactericidal effects, CP1 and OP2, which were synthesized based on two styrene-based copolymers P5 and P7 designed by the authors previous reports. The research presented physicochemical characterization of NPs to enable their mutual formulations as hydrogels for the development of antibacterial biomaterial and the new NPs were demonstrated to have antibacterial activities against several Gram-positive and Gram-negative multi-drug resistant pathogens. The research was properly designed and well conducted. The information is important and will contribute to the research field including biomaterials and medicine. Several comments should be addressed, which will strengthen the manuscript.

Questions

  1. The author explained that “some types of CNPs possessing primary ammonium groups have outperformed their tertiary and quaternary analogues in a lower level of toxicity” and “both CP1 and OP2 have neither quaternary nor permanently protonated ammonium groups but possess reversibly protonable primary amine groups”. This explanation can be interpreted as CP1 and OP2 having a low level of toxicity. However, there is a possibility that CP1 and OP2 could have toxicity to normal cells, as can be seen in the previously report which showed P5 and P7 exhibited to reduce NB cell viability in a concentration dependency. In addition, the authors also mentioned that dose dependent cytotoxicity experiments on human normal fibroblasts are undergoing to assess the possibility of CP1 and OP2 for clinical application in the discussion of this study. I agree with their discussion. However, cytotoxicity experiments are required in this study, not in the future, because the authors are seriously considering the usage of CP1 and OP2 as antibacterial biomaterials.

We thank the Reviewer for his comments and understand his curiosity concerning the results from cytotoxicity experiments, which we have available, but that as we have underlined in the present manuscript will be inserted in our next works. Precisely, we have ready for submission a first study where the self-forming gel obtained simply by dispersing CP1 in water was prepared and completely characterized by several analytical techniques. This work will contain the results from cytotoxicity experiments made with CP1 on human fibroblasts and the values of selectivity indices computed by those of DL50 which were up to 10 (as already reported in the Conclusions of the present manuscript). Currently, we are also studying the gel formulation of OP2 using CP1 as gelling agent. The preparation and characterization of such gel, as well as data of cytotoxicity of CP1 and OP2 separately (which we already have) and of CP1 and OP2 as gel formulation, will be the topics of a second work. Additionally, we predict you, that new MICs will be determined on the CP1-OP2-based gel and that both the latter and the CP1-based gels will be tested also for their antibiofilm properties. However, as abovementioned, we make kindly note the Reviewer, that in the Conclusions section, data concerning the selectivity indices for both CP1 and OP2 deriving from the obtained values of LD50 have been already reported in the present manuscript. Please, see lines 746-748.

Anyway, we explain to the Reviewer that to give data of cytotoxicity of CP1 and OP2 is out of scope of the present manuscript, where neither CP1 nor OP2 are formulated for clinical use. The scope of the present manuscript was the preparation and physicochemical characterization of new antibacterial/bactericidal cationic polymers potentially applicable in therapy after proper formulation. In this regard, we think that the biological data mandatory for this work are only those from microbiologic experiments. Surely, the Reviewer will agree with us that, microbiologically speaking, our work is more than complete. Additionally, it is no coincidence that we have chosen Nanomaterials rather than Pharmaceuticals, Pharmaceutics or Biomedicines as the journal to send our work to, as it is a journal that focuses more on the synthetic and characterizing aspect of the nano materials than on the biological one. Collectively, we think that the present work is already very complex and that the various microbiology experiments and the excellent microbiological results reported may be enough to make the work also pharmacologically attractive. We therefore kindly ask the Reviewer to be satisfied, and not to force us to further burden the work with the cytotoxicity data that will be communicated shortly. Moreover, we retain incorrect thinking of a possible cytotoxicity of CP1 and OP2 on normal human cells because similar compounds as P5 and P7 were cytotoxic against NB cells. As widely reported, cancer cells having surface more negative than that of normal cells are significantly more susceptible to the action of cationic macromolecules than non-cancer cells. However, to smooth out the fears of the Reviewer, we assure that, having demonstrated high selectivity indices, both CP1 and OP2 can be suggested for topical application and treat skin infections sustained by the most relevant strains reported in this work.

  1. The authors performed to the experiments of MICs and MBCs and the results showed remarkable antibacterial effects of CP1 and OP2. How about biofilm formation? I would like to know about the effects of them against biofilm formation.

We thank the Reviewer to have raised the global concern of biofilm eradication. We consider that biofilms are a very worrying forms of bacterial resistance, very difficult to be inhibited and/or disaggregated (when mature) which need resolutions. Bacterial and fungi biofilms formed on medical devices or medical implants are responsible of the most part of untreatable nosocomial infections, and new compounds with antibiofilm effects are urgently needed. In this regard, as previously explained, and due to the already complexity of data reported in this manuscript, the possible antibiofilm effects of CP1 and OP2 will be investigated when formulated as hydrogels in our next works. Another reason of this choice depends on the fact that in the form of hydrogels both polymers can be spread on any surfaces and can form a protective film by heating. We hope that such thin film could inhibit the attachment of bacteria and the whole organization of the biomass of the biofilm.

  1. The results of all MICs in this study were expressed reporting the modal value. To objectively prove the reliability of data, the authors should show all data of measured value including the measurements which were triplicate as a supplemental data.

We make kindly note to the Reviewer that it was not our choice to report the modal value. As detailed by EUCAST, the MIC value to be supplied is the concentration of the first well of a series of wells at increasing concentrations where no growth of bacteria was observed more frequently (modal value). Please, consider carefully the EUCAST. To give all observed values of MICs is microbiologically incorrect. In other words, according to the common practice in microbiology, MICs determination are usually carried out in triplicate or more and the numerical value that is reported as MIC corresponds to the values which has been obtained more frequently, that is a modal value.

Reviewer 2 Report

1. Except for DLS,  other methods should be used for measuring the nanopariticle parameters for example, SEM/TEM, ect.

2. In table 2, there is a huge difference of loading %, Why?

3. In table 6, the size of CP1 is larger than CP2, why the MIC concentration of CP1 is lower than CP2, please explain it .

4. In time killing curvem, the  variation of antibacterial activity of CP1 and CP2 should be explained.

5. In the introduction and methods, some reference should be added and discussed, for example,10.3967/bes2019.027; 10.1016/j.actbio.2021.11.010

Author Response

  1. Except for DLS, other methods should be used for measuring the nanopariticle parameters for example, SEM/TEM, ect.

We agree with the Reviewer and having available SEM micrographs obtained by SEM analyses made on CP1 recovered by lyophilization from its hydrogel formulation prepared using only water, whose ATR-FTIR was identical to that of original CP1, SEM analyses have been included for CP1 particles both in the Materials and Methods Section (lines 282-286) and in Results and Discussion one (lines 555-564), as confirmation of data by DLS.

  1. In table 2, there is a huge difference of loading %, Why?

Sorry, but where has the Reviewer detected values of loading % in Table 2 or in any other Table in our manuscript?  No loading % has been reported in the present study. As reported below, Table 2 includes the values of η rel measured for CP1 and OP2, the related η sp and [η], and the used values of a and K and the resulting Mr obtained by the Mark–Houwink equation.

.

Table 2. Values of η rel measured for CP1 and OP2, the related η sp and [η], the used values of a and K and the resulting Mr obtained by the Mark–Houwink equation Eq. (1).

Polymer

η rel (dL/g)

η sp

[η] (dL/g)

a*

K (dL/g)*

Mr

CP1

1.84 ± 0.02

0.84 ± 0.02

1.68 ± 0.02

0.93

0.0000250

157,306

OP2

1.26 ± 0.02

0.26 ± 0.02

0.52 ± 0.02

44,514

* Tabulated for poly 2-vinylpyridine in 0.1M Na formate at 30° C [18].

  1. In table 6, the size of CP1 is larger than CP2, why the MIC concentration of CP1 is lower than CP2, please explain it.

Even if we could simply respond that the MICs of CP1 are lower than those of OP2 because CP1 is more potent, some further considerations are mandatory.

First, we think that with “size” the Reviewer meant the molecular weight, since no size is reported in Table 6. Anyway, as the Reviewer can see, not always the MICs of CP1 are lower than those of OP2. Against A. baumannii and S. maltophilia OP2 showed MICs lower than those of CP1 (lines 642-648). Additionally, if the Reviewer considers the MIC expressed as µg/mL concentrations, the MICs of the two compounds are similar, and often those of OP2 are lower than those of CP1. The most part of MICs of CP1 are lower than those of OP2 if MICs if µM concentrations are considered, due to the higher molecular weight (Mr) of CP1.

  1. In time killing curvem, the variation of antibacterial activity of CP1 and CP2 should be explained.

We make kindly note to the Reviewer that time killing CURVES reported in the present manuscript show the bactericidal properties on CP1 and OP2 and not the antibacterial effects which has been demonstrated determining the MICs. Anyway, we are sure that no further explanation other than those already present in the text (lines 685-706) should be given on the difference in activity of the two compounds. We point out that both CP1 and OP2 are bactericidal on all tested strains. OP2 acts faster by exterminating all the colonies of bacteria, while CP1 leaves some colony when it is used on E. coli. All this is already said in the text. Please consider the lines 685-706.

  1. In the introduction and methods, some reference should be added and discussed, for example,10.3967/bes2019.027; 10.1016/j.actbio.2021.11.010

As requested by the Reviewer the reference corresponding to the doi: 10.1016/j.actbio.2021.11.010, has been including in the Introduction Section as Ref. 2 (line 35), and inserted in the References list (lines 769-781). The other reference suggested by the Reviewer is not suitable for our manuscript because concerns the synergistic effects of certain antifungal agents tested on C. albicans and does not deal with bacteria or cationic macromolecules.

Round 2

Reviewer 2 Report

the author have responsed the questions reasonably.